# Intermittency analysis of charged hadrons generated in Pb-Pb collisions at $\sqrt{s_{NN}}$= 2.76 TeV and 5.02 TeV using PYTHIA8/Angantyr

Salman K. Malik and Ramni Gupta

Department of Physics, University of Jammu, India
* salman.khurshid.malik@cern.ch

November 29, 2022

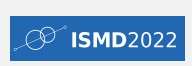

## Abstract

**Local density fluctuations are expected to scale as a universal power-law when the system approaches critical point. Such power-law fluctuations are studied within the framework of intermittency through the measurement of normalized factorial moments in $(\eta, \phi)$ phase space. Observations and results from the intermittency analysis performed for charged particles in Pb-Pb collisions using PYTHIA8/Angantyr at 2.76 TeV and 5.02 TeV are reported. We observe no scaling behaviour in the particle generation for any of the centrality studied in narrow $p_T$ bins. The scaling exponent $\nu$ shows no dependence on the centrality ranges.**

## 1 Introduction

Critical point and phase transition are being continously explored in heavy-ion collisions to understand quantum chromodynamics (QCD) phase structure. Lattice QCD predicts a crossover from hadronic matter to quark gluon phase (QGP) at $\mu_B = 0$ [1]. The first-order phase transition at large $\mu_B$, if exists, will end at a critical point [2]. However, the location of critical point and the nature of phase transition is highly uncertain. The rapid increase in the correlation length as the system approaches critical point gives rise to a system which is scale-invariant and fractal [3]. Additionally, the large density fluctuations form a self-similar structure in final state particles which can be studied within the framework of intermittency, which reveals itself as a power-law behaviour of Normalized Factorial Moments (NFM). An advantage of NFM is that they remove the associated statistical fluctuations and characterize non-statistical fluctuations connected with the dynamics of particle production [4]. In this paper, we report intermittency measurements in Angantyr/PYTHIA8 at $\sqrt{s_{NN}} = 2.76$ TeV and 5.02 TeV.

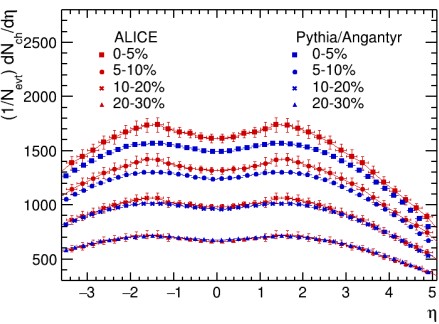 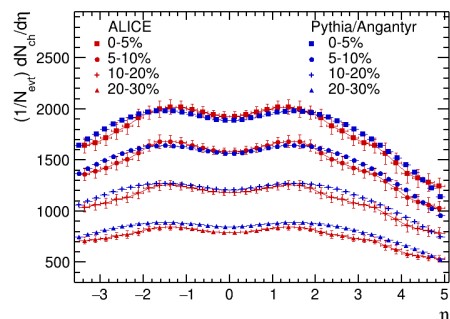

Figure 1: Charged particle pseudorapidity distribution for different centralities from Pb-Pb collisions using PYTHIA/Angantyr compared with that from ALICE at $\sqrt{s_{NN}} = 2.76$ TeV and 5.02 TeV [5,6].

## 2   Methodology

Intermittency [3,7] analysis has been performed in a two-dimensional $(\eta,\phi)$ phase space divided into $M \times M$ bins. The $q-$th order NFM are defined as:

$$F_q(M) = \frac{\frac{1}{N}\sum_{e=1}^{N}\frac{1}{M}\sum_{m=1}^{M}f_q(n_{me})}{\left(\frac{1}{N}\sum_{e=1}^{N}\frac{1}{M}\sum_{m=1}^{M}f_1(n_{me})\right)^q} \tag{1}$$

where $n_{me}$ is the number of particles in $m^{th}$ bin and $e^{th}$ event with $q$ being the order of the moment is an integer and is $\geq 2$. $f_q(n_{me}) = \prod_{j=0}^{q-1}(n_{me}-j)$.

For the systems approaching phase transition, multiplicities within the phase space are such that NFM exhibit power-law with decreasing bin size as:

$$F_q(M) \propto (M^D)^{\phi_q} \tag{2}$$

where $D$ (= 2 in this analysis) is the dimensionality of the phase space. This power-law scaling of NFM with the number of bins ($M^D$) is called *intermittency*. $\phi_q$ are intermittency indices. Intermittency studied within the realm of Ginzburg-Landau (GL) formalism [8]:

$$F_q(M) \propto F_2(M)^{\beta_q} \tag{3}$$

where, $\beta_q = \phi_q/\phi_2$. Equation 3 is called *F-scaling*. Intermittency index, $\phi_q$ and $\beta_q$ are different in that they depend on different critical parameters of the system. This implies that even if Equation 2 dependence (M-scaling) is absent in a system, F-scaling can still be independently analyzed. $\beta_q$ is described by *scaling exponent*, $\nu$:

$$\beta_q \propto (q-1)^{\nu} \tag{4}$$

Scaling exponent ($\nu$) is independent of the parameters of the system. Its value is predicted to be 1.304 in Ginzburg-Landau theory for second-order phase transition and 1.0 from the 2D Ising model calculations [8] for critical fluctuations.

Angantyr is the heavy-ion generator of PYTHIA8 [9]. It doesn't assume a hot thermalised medium but it rather extrapolates pp dynamics to heavy-ion collisions. Angantyr gives a good description of final state particles in pA and AA collisions [9]. Figure 1 shows the charged particles pseudorapidity distribution for the events used in this analysis compared with ALICE data [5,6].

## 3 Observations

Two million events generated using PYTHIA+Angantyr for Pb-Pb collisions at $\sqrt{s_{NN}} = 2.76$ TeV and 1M at 5.02 TeV have been analyzed in the midrapidity region ($|\eta| \leq 0.8$) with centralities (the centrality is defined by $\sum E_t$ of the events) 0-5%, 5-10%, 10-20%, 20-30% for many differing width $p_T$ bins out of which the results for $0.4 \leq p_T$ (GeV/$c$) $\leq 1.0$ are shown. NFM are calculated for $q = 2,3,4$ and 5. Number of phase space bins, $M$ is taken from 4 to 84 in the intervals of 2.

Behaviour of NFM ($F_q's$) with number of bins $M$ ($M$-scaling) is given in Figure 2a, and Figure 2b shows $F_q's$ ($q = 3,4,5$) dependence on second-order NFM ($F_2$). It is observed that $F_q's$ are independent of $M$, and thus $M$-scaling is absent. However, a weak dependence of $F_q's$ on $F_2$ is observed and with scaling exponent ($\nu$) = $1.945 \pm 0.112$ (Figure 2c). But '$\nu$' gives quantitative characterization of spatial fluctuations of particles generated in Angantyr. Figure 3 gives the scaling exponent for $0.4 \leq p_T$ (GeV/$c$) $\leq 1.0$ bin studied for different centrality bins and $\gg 1.304$ value predicted by GL formalism for second order phase transition.

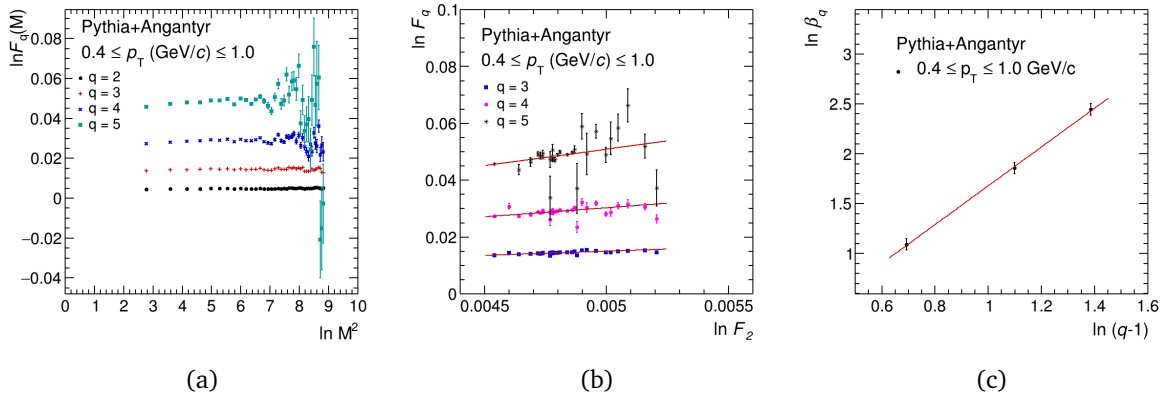

(a)                                      (b)                                      (c)

Figure 2: $\ln F_q$ dependence on (a) $\ln M^2$ and (b) $\ln F_2$, (c) $\ln \beta_q$ vs $\ln(q-1)$ plot to obtain scaling exponent ($\nu$) for the $p_T$ bin $0.4 \leq p_T$ (GeV/$c$) $\leq 1.0$.

## 4 Conclusions

Investigations on the intermittency analysis for Pb-Pb collisions at 2.76 and 5.02 TeV within PYTHIA+Angantyr have been reported. It is concluded that no M-scaling is present in the particle generation particularly in narrow $p_T$ bins is absent and no self-similarity in fluctuations. Hence, scale-invariant fluctuations are completely absent. In case of wide $p_T$ bins, F-scaling is observed and the value of $\nu$ is 1.7-1.9 for different centralities. In Angantyr, the value of $\nu$ is greater than the value put forward in GL theory. Within statistical errors, $\nu$ is independent of the centrality ranges.

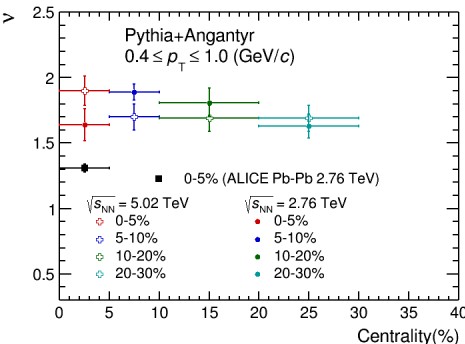

Figure 3: Centrality dependence of $\nu$ for Pb-Pb collisions at $\sqrt{s_{NN}}$ = 2.76 TeV and 5.02 TeV. Scaling exponent for same $p_T$ bin from ALICE experiment at 2.76 TeV also shown [10].

## Acknowledgements

We are thankful to the Angantyr developers and particularly to Harsh Shah for his help regrading compatibility with ROOT and centrality calculations. The authors are also thankful to RUSA2.0 to University of Jammu by Ministry of Education, India for partial support for computing resources.

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
