# Peer review of "Intermittency analysis of charged hadrons generated in Pb-Pb collisions at $\sqrt{s_{NN}}$= 2.76 TeV and 5.02 TeV using PYTHIA8/Angantyr"

_SciPost Physics Proceedings_

## Round 1 · Referee Report · Anonymous (Referee 1) · 2022-11-16

Strengths

1) The paper is easy to read. It manages to describe the essential features of the work using only 4 compact formulas without overwhelming the reader with a lot of details.

Report

The authors study local density fluctuations in Pythia/Angantyr model by using normalized factorial moments. The authors do not see signs of scale invariant fluctuations. The authors also extract scaling exponent, which has been predicted for a second order phase transition using Ginzburg-Landau theory. The extracted exponent does not agree with the value predicted for the phase transition. This is an expected result, since Pythia/Angantyr does not assume formation of thermalized plasma. Thus there is no a priori reason to expect to see signs of phase transition in Pythia/Angantyr. The significance of the result is that a model without a phase transition can not reproduce the scaling exponent which is expected of a phase transition. Thus increasing confidence in this observable. I recommend publication.

Requested changes

While reading, I have noticed a few typos etc. that the authors may want to fix for the final version:.
Typos:
-"It’s value is predicted to be"-> Its value is predicted to be
-generaton -> generation
-crtitical -> critical
-"different centrality bins and >> 1.304" I think you should use the latex command \gg here instead of ">>".

Figures:
-Fig. 3 the legend for (ALICE Pb-Pb...) seems to extend beyond the boundaries of the figure.

  • validity: good
  • significance: good
  • originality: good
  • clarity: high
  • formatting: reasonable
  • grammar: reasonable

Author:  Salman Malik  on 2022-11-29  [id 3086]

(in reply to Report 1 on 2022-11-16)

Thank you so much for the comments. I have resubmitted with the mentioned changes.

---

## Editorial Decision

editorial_decision: